# Deconstructing Fat to Reverse Radiation Induced Soft Tissue Fibrosis

**DOI:** 10.3390/bioengineering10060742

**Published:** 2023-06-20

**Authors:** Hannes Prescher, Jill R. Froimson, Summer E. Hanson

**Affiliations:** Section of Plastic & Reconstructive Surgery, University of Chicago Medical Center, Chicago, IL 60615, USA

**Keywords:** radiation-induced fibrosis, fat grafting, adipose-derived stem cells, biological scaffolds, regenerative medicine, stromal vascular fraction, exosomes

## Abstract

Adipose tissue is composed of a collection of cells with valuable structural and regenerative function. Taken as an autologous graft, these cells can be used to address soft tissue defects and irregularities, while also providing a reparative effect on the surrounding tissues. Adipose-derived stem or stromal cells are primarily responsible for this regenerative effect through direct differentiation into native cells and via secretion of numerous growth factors and cytokines that stimulate angiogenesis and disrupt pro-inflammatory pathways. Separating adipose tissue into its component parts, i.e., cells, scaffolds and proteins, has provided new regenerative therapies for skin and soft tissue pathology, including that resulting from radiation. Recent studies in both animal models and clinical trials have demonstrated the ability of autologous fat grafting to reverse radiation induced skin fibrosis. An improved understanding of the complex pathologic mechanism of RIF has allowed researchers to harness the specific function of the ASCs to engineer enriched fat graft constructs to improve the therapeutic effect of AFG.

## 1. Introduction

Autologous fat transfer or grafting is commonly used in reconstructive surgery to correct soft tissue defects created by oncologic resection [1]. It has been observed clinically to provide a rejuvenating effect on the overlying dermal architecture of the recipient bed though this mechanism is not fully understood. Scientists interested in the regenerative role of fat grafting attribute this effect, in part, to the action of mesenchymal progenitor cells which can be isolated from adipose tissue [2]. Over recent decades, processing techniques have made it possible to deconstruct fat to isolate such adipose-derived stem or stromal cells (ASC) from the stromal vascular fraction (SVF) of lipoaspirate or purify the stromal matrix to use as a biological scaffold [3]. 

Radiation induced fibrosis (RIF) is a pathologic consequence of radiation therapy and is characterized by dermal thickening, reduced tissue elasticity and microvascular obliteration [4]. Clinically, this manifests as pain, disfigurement and functional compromise of affected tissues with high patient morbidity and decrease in quality of life. While the mainstay of treatment for fibrotic, irradiated tissue includes direct excision and soft tissue reconstruction, fat grafting and transfer of ASCs have shown promising results in the reversal of RIF. Although this mechanism has not been fully elucidated, ASCs are thought to orchestrate tissue regeneration primarily via secretion of cytokines and growth factors with angiogenic, adipogenic and antifibrotic effects [5]. More recently, we see a similar benefit from delivery of the decellularized adipose matrix (DAM) in repairing the dermal architecture after ionizing radiation. Current research is focused on identifying the various molecular pathways that contribute to RIF as potential targets for treatment. This has important implications for engineering fat grafts, as these can be enriched with specific subpopulations of ASCs to directly interfere with specific molecular modulators of the pathologically activated profibrotic cascade. Cellular preconditioning of these subpopulation of ASCs with target specific growth factors in culture medium, and optimal packaging of the grafts can promote graft survival potential at the recipient site and improve outcomes. The purpose of this review is to highlight the state-of-the-art bioengineering strategies being employed to offer a novel approach to an incurable condition. 

## 2. Structural Components of Fat Grafts

Adipose tissue is composed primarily of mature adipocytes. Adipocytes may be either uni- or multi-loculated and function in thermoregulation, mechanical cushioning and energy storage [6]. Grafted tissue is typically harvested from the subcutaneous layer of the abdomen, flanks or thighs and contains adipocytes along with a heterogeneous collection of cells called the SVF. This includes MSCs/ASCs, preadipocytes, fibroblasts, vascular smooth muscle, endothelial cells, macrophages and lymphocytes [7,8]. This portion of the stroma can be further processed to isolate the ASCs based on specific surface markers, which can then be culture-expanded for therapeutic use.

There are multiple factors that can affect the structure and cellular composition of a fat graft including obesity, age and chronic diseases. ASCs isolated from elderly and obese patients have reduced function and differentiation potential. They have reduced angiogenic potential and lower expression of multipotency associated genes [9,10,11,12]. The method of fat harvest can also have a significant effect on the structure and cellular composition of the graft. When comparing aspirated to excised adipose tissue, Eto et al. showed that fat obtained via liposuction (i.e., aspirated) had a reduced capillary network and lacked large vascular structures [13]. Critically, the yield of adipose derived stromal cells containing stem cells was approximately one-half in aspirated compared to excised fat tissue. Fat processing techniques, purification of the lipoaspirate, recipient site preparation and the delivery method of the harvested fat to the recipient bed [14,15] are equally important. While there remains great variability in reported techniques, experts have reached consensus on the use of larger harvesting and grafting cannulas and slow injection speed to limit cell damage and maintain the cellular structure of the graft. 

The structure of the fat graft is important for subsequent injection and engraftment. Like any graft, adipocytes rely on the surrounding tissues for survival after being injected into a recipient site. Initially, the graft exchanges nutrients with the surrounding tissues via cellular diffusion (imbibition), which is followed by the growth of new microvasculature (neovascularization) and the establishment of a more robust gas and nutrient exchange network. Adipocytes are particularly sensitive to hypoxia, and owing to its spherical structure, the graft often undergoes some degree of central necrosis. Maintaining the capillary network shown to run alongside adipocytes is therefore critical for engraftment, while the SVF carries angiogenic and adipogenic factors that regenerate the microenvironment of the recipient tissue bed [16]. 

## 3. Clinical Applications 

Autologous fat grafts have long been used as filler to correct soft tissue defects. In cosmetic surgery, fat grafting is commonly used for facial rejuvenation to correct asymmetries and to augment age-related soft tissue atrophy. Fat grafts are also used both in body contouring surgery and in breast surgery for augmentation or for volumetric restoration of an oncologic defect [17,18]. The emphasis has historically been on the volumetric component of the fat graft with early recognition of high variability in volumetric graft retention. However, only relatively recently have advances in the understanding of its cellular composition, and the role of adipose-derived stem cells led to increasing application of fat grafting in regenerative medicine. In particular, fat grafting has been shown to improve chronic wound healing and promote scar remodeling [7,19,20]. Fat grafting has also yielded promising results in the treatment of diabetic ulcers [21,22], as researchers demonstrated significantly faster healing in wounds treated with allogeneic ASC sheets compared to standard wound care [21]. 

The etiology of chronic wounds is highly variable and influenced by both internal factors (age, comorbidities, and systemic therapies) and external factors (radiation, trauma, and burns). However, there are commonalities in the microenvironment of chronic wounds including elevated levels of proinflammatory cytokines and reactive oxygen species, decreased growth factors and stem cells, dysregulation of the extracellular matrix and tissue hypoxia [23]. Engineered fat grafts have emerged as a unique delivery modality of cells that can disrupt these pathologic pathways and thereby provide a regenerative therapy for a number of chronic inflammatory conditions. Radiation induced soft tissue injury is one such chronic condition.

## 4. Radiation Induced Fibrosis: Mechanism 

Radiation induced fibrosis (RIF) is the pathologic consequence of soft tissue exposure to ionizing radiation. It is both an acute and chronic process that leads first to destruction of highly proliferative cell lineages including basal keratinocytes in the skin [24] and later to a chronically upregulated pro-inflammatory cascade resulting in soft tissue fibrosis and scarring [25]. In the acute phase, this process manifests clinically as radiodermatitis, characterized by edema, erythema, epidermal thickening, skin sloughing, pruritis and desquamation [4,26]. The chronic phase of RIF is triggered by the release of tissue growth factor beta 1 (TGF-β1) [27] in affected tissue from direct damage to resident cells. Increased expression of TGF-β1 disrupts extracellular matrix (ECM) homeostasis by increasing deposition of proteins while inhibiting ECM proteases that help to break down excess matrix. It also stimulates the differentiation of fibroblasts into myofibroblasts, which results in increased production of collagen, fibronectin and proteoglycans [28]. 

Sustained aberrant activity of these cell populations over time leads to remodeling of collagen into rigid, inelastic bundles. This process is paralleled by distinct changes to the vascular architecture of the tissues. The initial response to RT is marked by increased vascular permeability, while chronic inflammation leads to the formation of fibrin clots, intravascular thrombosis and hypoperfusion of tissues [29]. The attendant tissue hypoxia stimulates further production of TGF-β1, which promotes differentiation of fibroblasts, increases collagen production and sustains the profibrotic microenvironment of the irradiated tissues in a pathologic cycle [30]. 

## 5. Radiation Induced Fibrosis: Clinical Impact

Radiation therapy (RT) alone or in conjunction with surgery has become instrumental in treating a variety of cancers and has been shown to improve overall survival rates [31,32,33]. While the precision of RT delivery has improved over time, it continues to create significant collateral damage to the surrounding healthy tissues. Clinically, this manifests as dermal thickening, decreased tissue elasticity, fat atrophy and contracture of the affected tissues leading to limited mobility, delayed wound healing and radionecrosis with ulceration [2,30]. Significant functional impairments can result, including dysphagia, trismus, xerostomia, alopecia and facial disfigurement in patients with head and neck cancer [34]. These side effects can be particularly debilitating as the head and neck region plays vital functional and aesthetic roles and is central to a person’s sense of self. RT is a primary intervention for many patients with advanced oropharyngeal cancers [33], and even low dose radiation can result in inflammation of the oropharyngeal mucosa, painful swallowing, hoarseness and nausea and vomiting [35]. The inability to swallow can lead to inadequate nutrition and hydration and ultimately to weight loss and failure to thrive. These complications cannot only interfere with and delay further treatment but also represent a significant reduction in quality of life for the patient. In a recent review of patient reported outcomes in cancer survivors with RIF, patients complained particularly of functional impairment, appearance related distress and loss of emotional wellbeing as a result of their treatment [36].

Significantly, these side effects have been shown to persist even after treatment completion. In a study of 72 patients treated with RT for nasopharyngeal carcinoma, functional outcomes were shown to initially improve at 3 months after completion of therapy. However, this was followed by a subsequent decline years post-treatment with advancing dysphagia, aspiration, trismus, muscle spasm and hypoglossal nerve palsy [37]. The probability of a poor outcome increased with time. These clinical findings are consistent with the acute and chronic phase of RIF and demonstrate the high patient morbidity associated with chronic, uninhibited tissue fibrosis. 

Similar effects are observed in patients with breast cancer. While surgical excision remains the predominant treatment modality for breast cancer, RT in both the neoadjuvant and adjuvant setting is often used to decrease tumor size and the risk of cancer recurrence [32,38,39,40]. It is estimated that by 2030 in the United States alone there will be greater than 2 million radiation-treated breast cancer survivors [31]. RT protocols may vary significantly, leading to a spectrum of clinical severity in associated symptoms. RIF commonly presents as pain, tissue contracture, ulceration and atrophy of breast tissues. These soft tissue changes have a significant impact on breast reconstruction. Alloplastic reconstruction with breast implants remains the most common form of reconstruction, and RT delivered pre- and post-mastectomy has been associated with greater surgical complications and worse aesthetic outcomes [41,42,43,44]. Irradiated breasts are more susceptible to wound infection, capsular contracture and implant exposure and subsequent loss [43]. Patients undergoing RT also required more frequent reoperation and experienced delay in achieving reconstruction. Ultimately, RT can have a significant impact on a patient’s quality of life [45,46] by decreasing physical functioning and causing anxiety, lower body image and an increase in pain [47,48,49]. 

Radiation of lymphatic tissue in the axilla may also lead to lymph node dysfunction and result in upper extremity lymphedema [50]. Recent studies showed equivalent overall and disease-free survival and locoregional control between patients with a positive sentinel axillary lymph node treated with complete axillary dissection versus axillary radiation [51]. The rates of upper extremity dysfunction and lymphedema may therefore be expected to increase in the future. Symptoms related to lymphedema including repeated episodes of cellulitis, skin breakdown and functional impairment can emerge both in the acute and chronic phase of RIF as progressive fibrosis and contracture of the axillary lymph node basin impairs lymphatic drainage over time.

## 6. Role of Fat Grafting in Treating Radiation Induced Fibrosis

A central tenet of reconstruction is to “replace like with like” meaning when faced with a defect or deformity the area to be reconstructed should be conducted so with a tissue source that has similar size, texture, color, rigidity and other characteristics. For example, a breast is best reconstructed with skin and fat tissue from the abdomen, while the jaw is best replaced with a composite flap including an expendable bone like a portion of the fibula or scapula. As an alternative to these costly, complex, potentially morbid operations, tissue engineering strategies seek to repair and regenerate the damaged tissue on a molecular level. One such approach is founded in autologous fat transfer (AFT). Structurally, fat grafting is an autologous filler to improve contour irregularities and soft tissue atrophy of radiation-induced skin contracture. Molecularly, there is evidence to suggest that the graft can directly interfere with the chronic inflammatory cascade of the affected tissues. Recent preclinical and clinical studies in autologous fat grafting have shown promising results in improving previously irradiated tissue fields.

Other studies have shown that fat grafting decreases dermal thickness and collagen content, while increasing vascular density [52,53]. In these studies, samples of fat-grafted tissues revealed a critical downregulation of the profibrotic TGF-β1/Smad3 signaling pathways. Lindegren et al. further demonstrated a reversal of dysregulated gene expression in irradiated tissue following fat grafting [54]. The authors found a significant response in genes involved in both the hypoxia pathway as well as the interferon-gamma pathway, which is intimately involved in inflammation and fibrosis. Gene expression profiles showed return to nearly normal levels after fat grafting, which correlated with normalization of the microcirculation seen on electron microscopy.

These findings have important clinical implications. Rigotti et al. were the first to demonstrate symptomatic and visible improvement following fat grafting in irradiated breast cancer patients [55]. Examining 120 patients with severe RIF following RT for breast cancer, they showed dramatic changes in the dermal architecture of the irradiated tissue with increased hydration and capillary density after a single treatment with AFT. These histologic changes correlated with clinical improvement. Other studies have demonstrated the positive impact of AFT on both the functional and aesthetic outcomes in breast reconstruction [56]. AFT has also been used prophylactically to reverse the pathologic effects of RIF in breast tissues prior to breast reconstruction [57,58,59]. Ribuffo et al. demonstrated that two separate AFT procedures to the breasts 6 weeks after RT and 3 months prior to completion of implant-based breast reconstruction dramatically reduced the rates of skin ulceration and implant exposure [58]. The shape and symmetry of the reconstructed breasts were found to be significantly better in the AFT-treated patients. AFT has also been shown to create significant and sustained improvements in postmastectomy pain, allowing patients to be weaned from narcotic pain medications [60,61,62,63]. These findings have been confirmed in other studies without an increase in the risk of cancer recurrence following injection of ASCs into surgical sites that previously contained cancerous cells [64,65]. Most importantly, fat grafting has been shown to improve the outcome in breast reconstruction independent of reconstruction type and increase patient satisfaction and quality of life [66].

Likewise, AFT has been shown to produce significant functional and aesthetic improvements in patients with RIF of the head and neck region following RT. Patients expressed high satisfaction with the volumetric restoration procedure as 77.5% and 89.2% reported aesthetic and functional improvements [67]. Specifically, AFT improved neck mobility, jaw mobility and laryngeal compliance leading to restoration of speech and swallowing function [68,69]. It has also been shown to reduce xerostomia [70] and improve skin texture and facial appearance [71]. These clinical effects of AFT are significant as improvements in swallow function can lead to an increase in PO intake, lower reliance on gastrostomy feeding and dramatically improve the quality of life of these patients. 

## 7. Engineering Strategies to Optimize Regenerative Potential of Adipose Tissue

### 7.1. Cell-Assisted Lipotransfer

In spite of the clinical success achieved by fat grafting, delivering autologous fat into a hostile, irradiated tissue field remains a challenge. Adipocytes are particularly sensitive to hypoxia, which is a hallmark of irradiated tissues. To combat this challenge, surgical technique and product development—the future of fat grafting—have focused on engineering fat grafts with augmented survival potential by manipulating the composition of the grafts [72]. Figure 1 illustrates the various concepts behind engineered adipose therapeutics in regenerative medicine [73]. Cell-assisted lipotransfer (CAL) was introduced by Matsumoto et al. in 2006 in an effort to optimize the survival of adipocytes in the chronically fibrotic, hypoxic microenvironment [74]. The authors demonstrated that augmentation of a fat graft with adipose-derived stem cells could increase the volumetric retention of the graft in the irradiated tissue. Others confirmed these findings [72,75,76,77] and showed that the ASCs are responsible for the phenotypic changes observed in the recipient tissue bed [76].

### 7.2. Adipose-Derived Stem/Stromal Cells

Adipose-derived stromal cells (ASCs) have extensive proliferative potential and comprise up to 3–10% of the stromal vascular fraction of adipose tissue [78]. When isolated and injected into animal models, the cells were shown to promote wound healing [79,80,81,82], soft tissue healing [83], functional recovery of limb contractures [84,85], recovery of radiation-induced intestinal injuries [86] and improvement in salivary gland function [87]. In a clinical study including five patients with radiation-induced skin injury, injection of autogenous stromal vascular fraction into the injured tissue resulted in improved wound healing and pain relief [78], though it is important to remember that the SVF carries a heterogenous mix of cells in addition to ASCs. 

The precise mechanism by which these cells induce changes in the tissue microvasculature, collagen density and dermal architecture, however, continues to be debated. The current model proposes several pathways. ASCs have been shown to have trilineage differentiation capacity for bone, cartilage and fat [88]. Several studies have demonstrated direct differentiation of ASCs into angiogenic and adipogenic cell lineages to promote formation of new blood vessels and replace adipocytes [79,80]. Nie et al. observed spontaneous site-specific differentiation of ACSs into epithelial and endothelial cells via immunofluorescent analysis of ACSs co-stained with pan-cytokeratin and CD31 [80].

ASCs are also known to orchestrate tissue regeneration via secretion of paracrine growth factors including vascular endothelial growth factor (VEGF), tissue growth factor beta 3 (TGF-β3), basic fibroblast growth factor (bFGF) and hepatocyte growth factor (HGF) [89]. These cytokines and growth factors promote neovascularization from native tissue cells, while inhibiting fibroblast proliferation by decreasing TGF-β1 and promoting collagen remodeling by increasing TGF-β3 [90]. Ejaz et al. demonstrated these effects using histological and molecular analysis in a mouse model of RIF [89]. The phenotypic improvements in epithelial thickness, collagen deposition and limb excursion compared to non-irradiated controls were correlated with downregulation of fibrotic gene expression and recruitment of bone marrow cells to the irradiated site.

While promoting neovascularization through both direct differentiation and paracrine signaling, ASCs also release anti-inflammatory and anti-apoptotic cytokines that reduce the effects of tissue hypoxia on engrafted adipocytes [91,92,93]. Increased levels of interleukin-10 (IL-10) and reduction in interferon-gamma (INF-γ) help to modulate the chronically activated inflammatory response and improve survival of viable adipocytes. 

More recent work has highlighted the role of distinct subpopulations of stem cells within the SVF [94,95]. Borrelli et al. found that CD74+ enriched stromal cells had increased expression of HGF, FGF2 and TGF-β3 compared to CD74− or unsorted ASCs [95]. This correlated with greater reduction in dermal thickness and stiffness as well as collagen content. Other studies have identified surface markers on ASCs that confer specific regenerative roles. For example, bone morphogenetic protein receptor-1A enhances the capacity for adipogenesis [96], while endosialin (CD248) characterizes ACSs with angiogenic potential [97]. The specific regenerative roles of the various ASC surface markers continue to be elucidated, and researchers have levied this improved understanding to engineer augmented or enriched autologous fat grafts. Using a murine model of RIF, Deleon et al. showed that enrichment of fat grafts with CD34 + CD146 + ASCs enhanced fat graft vascularization and retention and significantly improved the quality of the radiation-injured soft tissues [98].

As our understanding of the pathologic mechanism of RIF improves, fat grafts can be engineered with increasing functional specificity to target key elements of the inflammatory cascade. Several strategies have emerged including preconditioning cell cultures with Vitamin E [99] or deferoxamine [100] to improve volume retention and upregulate VEGF expression, respectively. Researchers have also found that ASCs cultured in an endothelial growth medium (EGM) enhanced proliferative and multilineage potential. When these cell cultures (ASC-EGM) were then injected into immunodeficient mice, they produced significantly greater functional vasculature compared to ASCs grown in conventional media [101]. Co-culturing ASCs with fibroblasts has also been shown to promote mutual proliferation and improve adipogenic differentiation, hemangioendothelial differentiation and proliferation [102]. Significantly, in animal experiments, the authors also found that the autografted adipose group combined with both ASCs and fibroblasts had the lowest rate of oily cysts and the best volumetric retention rate.

Genetic modification of ASCs with VEGF-modified mRNA has also been shown to promote angiogenesis and long-term graft survival in a fat graft transplantation model [103]. The use of such genetic engineering with VEGF in a former tumor bed, even in the setting of completed radiation or fibrosis, carries the concern of oncogenic potential and may have limited utility in cancer reconstruction; however, this work demonstrates the capability of genetic engineering and could be applicable to other immunomodulators of interest.

### 7.3. Decellularized Adipose Matrix 

Post-harvest processing of fat grafts has led to the development of fat allografts in the form of decellularized adipose matrixes (DAM). DAMs are allografts that are derived from disposed human lipoaspirate and processed via physical, chemical and enzymatic purification techniques to create decellularized scaffolds that retain the complex macromolecular architecture of the adipose cells and their paracrine function. DAM grafts have been shown to promote adipogenesis and angiogenesis and are effective as soft tissue fillers and at treating RIF [104,105,106]. The ability of DAM allografts to induce a regenerative effect on tissues is encouraging as it eliminates the need for harvesting, which may be a limiting factor in some patients. 

### 7.4. Exosomes

Exosomes are extracellular vesicles enclosed by lipid membranes that can transport a variety of cellular components and fuse with target cells [107]. They can be isolated from lipoaspirate and retain their source cell characteristics, are easy to store and are immunogenic, making them ideal candidates for cell-based therapy. Exosomes carry a variety of bioactive substances including proteins, cytokines, microRNA and lipids that are involved in regulating a number of physiologic and pathologic processes. They are able to incorporate into a target cell via endocytosis and regulate its gene expression to restore its native function [108]. This has important implications in the treatment of chronic wounds and RIF, where proinflammatory, profibrotic and anti-angiogenic cell mediators are chronically upregulated. Studies have shown that injection of exosomes derived from ASCs can promote angiogenesis and adipogenesis, while modulating the deposition of ECM [109]. Exosomes also have important modulating function on fibroblasts, a cell population critical for wound healing. Hu et al. showed that exosomes can be internalized by fibroblasts and stimulate cell migration, proliferation and collagen synthesis in a dose-dependent manner [110]. These findings were confirmed by other studies that further demonstrated the upregulation of the Wnt/β-catenin signaling pathway, which plays an important role in the proliferative phase of wound healing [111,112,113,114,115]. Exosomes have also been shown to significantly reduce the effects of radiation-induced lung and intestinal injuries [116,117]. 

## 8. Conclusions

Radiation-induced fibrosis occurs through a complex molecular mechanism that causes chronic tissue damage and high patient morbidity. Currently, treatment options seek to replace the damaged tissue. Autologous fat grafting has been shown to improve soft tissue quality, and adipose-derived stromal cells are effective in combating this pathologic process and regenerating affected tissue. Molecular engineering of fat grafts with specific population of cells can enhance their angiogenic and adipogenic effects and lead to improved healing. Exosomes carrying microRNA derived from ASCs can alter gene expression in affected cells and directly alter the cell’s function. Clinical trials are required to demonstrate the efficacy and safety of these regenerative treatment modalities in post-oncologic or traumatic reconstruction. 

## Figures and Tables

**Figure 1 bioengineering-10-00742-f001:**
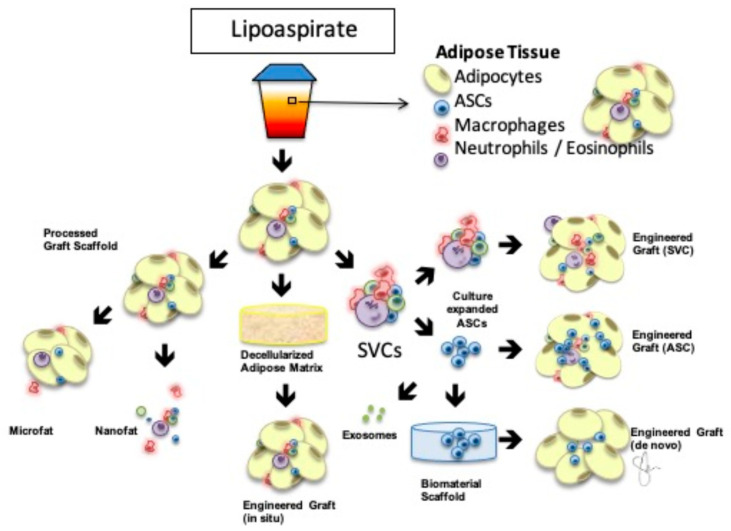
Engineered AdipoSe Therapeutics in Regeneration (EASTR) Concept Adipose-based therapeutics in regenerative medicine begin with processed lipoaspirate. Adipose tissue contains multiple stromal vascular fraction cells (SVCs) illustrated in the upper right, and the resulting graft is a natural tissue scaffold that can be further deconstructed for ultra-purified grafting as in microfat or nanofat or supplemented with additional cells including SVCs or a pure population of adipose-derived stem cells (ASCs). Culture-expanded ASCs could be encapsulated in a biocompatible, biomaterial scaffold for de novo engineered grafts or used to develop exosomes based on their paracrine effect, removing the cellular component out of the construct. Finally, decellularized adipose matrix can be delivered as an off-the-shelf allograft for in situ grafting without the additional donor site. Modified with permission [73].

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
