# Peer review of "Deconstructing Fat to Reverse Radiation Induced Soft Tissue Fibrosis"

_bioengineering, 2023, doi:10.3390/bioengineering10060742_

Round 1

Reviewer 1 Report

Authors have provided a brief summary of research regarding the therapeutic potential of adipose tissue transplantation on radiation-induced tissue fibrosis. This review will also contribute to an advanced understanding of the pathophysiology of radiation-induced fibrosis.  

Author Response

The authors appreciate the review of our manuscript and the word count has been addressed to make the manuscript more robust. SEH

Reviewer 2 Report

General Comments

This is a well written and reasonably well cited review paper that provides a clear and  concise evaluation of the status of adipose-derived materials for repair of radiation injuries of the skin and other tissues.  The authors consider the current state of knowledge as well as directions for future research.  The limitations of the field are evaluated as well.  It might be informative to highlight what the authors consider the most important “next experiments” to advance the field in their final concluding remarks.   

Specific Comments

Ln 29, 159.  Change acronym ASC to adipose derived stromal/stem cells (not just stem cells or stromal cells)

Ln 75.  Begin to use the acronym ASC instead of spelling out adipose derived stem cells.

Ln 213, Section 6.3.  There is a considerable body of literature relating to decellularized adipose matrix.  Further primary studies characterizing these materials in vitro and in vivo should be cited in addition to references 60-62.

Author Response

The authors appreciate the review of this manuscript. The changes have been made in the body to address the comments raised by the reviewer. The additional text brings the word count to 3958 excluding the headings, keywords and abstract. Please contact me with any additional questions or concerns. SEH